# Extracting Nonlinear Symmetries
# From Trained Neural Networks on Dynamics Data

**Yoh-ichi Mototake**[*]
Graduate School of Social Data Science
Hitotsubashi University
Tokyo, 186-8601
y.mototake@r.hit-u.ac.jp

## Abstract

To support scientists who are developing the reduced model of complex physics systems, we propose a method for extracting interpretable physics information from a deep neural network (DNN) trained on time series data of a physics system. Specifically, we propose a framework for estimating the hidden nonlinear symmetries of a system from a DNN trained on time series data that can be regarded as a finite-degree-of-freedom classical Hamiltonian dynamical system. Our proposed framework can estimate the nonlinear symmetries corresponding to the Laplace–Lunge–Renz vector, a conservation value that keeps the long-axis direction of the elliptical motion of a planet constant, and visualize its Lie manifold.

## 1 Introduction

One of the central roles in scientific activities is understanding large-scale complex systems through their reduced models. Some complex systems are modeled as low-dimensional canonical dynamical systems. For instance, reduced models have been developed for large-scale collective motion systems, which are a type of large-scale complex system with order, such as plasma, acoustic waves, or vortex systems [1, 2, 3, 4, 5]. To develop these reduced models, collective coordinates have been introduced, such as the Fourier basis of a density or charge distribution [1, 2, 3, 4], or a vortex feature space [5]. Then, a Hamiltonian that describes the coarse-grained properties of a dynamical system is derived. Thus, to develop a reduced model, it is necessary to introduce collective coordinates and derive the Hamiltonian in those coordinates. The obtained Hamiltonian is then verified by confirming that it can reconstruct the properties of the phenomena analyzed. This approach relies heavily on the physical insights of physicists and may not work for modeling a dynamical system that features a more complicated structure. One example is the collective motion of living things such as fish or birds; such systems frequently have stable but very complicated patterns in a metastable state [6, 7].

The problem we are considering here is how to infer a reduced model using machine learning methods. As mentioned earlier, this involves solving two problems: estimating a coordinate system and constructing a reduced model within that coordinate system. One way to solve these problems is to construct a Hamiltonian based on a given coordinate system and search for a coordinate system that improves the model. Several machine learning methods have been developed for inferring the Hamiltonian from a time-series dataset [8, 9, 10, 11]. These methods can be roughly divided into two types. In the first type, the Hamiltonian is inferred by regressing the data with an explicit function, such as the linear sum of multiple basis functions [8]. However, when inferring a reduced model that consists of complicated unknown basis functions, this method only infers an approximated reduced model using an approximated function, such as a polynomial function. In the second type, a Hamiltonian is modeled using deep learning techniques [9, 10, 11]. In this case, an explicit function

---

[*]https://mototakelab.github.io/mototake.github.io

used in the first type is not required. Based on these machine learning methods, the search for the coordinate system could be performed using statistical criteria such as the prediction error.

There are inherent difficulties in building a reduced model using a machine learning approach. Such an approach finds a Hamiltonian that has properties that only hold for the given data. Historically, physicists have achieved great success in constructing reduced models by abstracting knowledge obtained from observational data and building universal models that can explain various physical phenomena, not just the given data. For example, in thermodynamics, Gibbs linked a reduced model that describes the molecular motion of a gas to chemical reaction theory [12, 13]. This is one of the most successful uses of a reduced model. In other words, a good reduced model and a good coordinate system mean that the performance is high not only for the given data.

To achieve a successful reduced model, it is important to interpret the knowledge obtained during data analysis and develop a model that can be applied to different phenomena by combining explicit and implicit knowledge of physics. In general, an inferred Hamiltonian modeled by deep neural networks (DNNs) is difficult to interpret because DNNs are models with enormous degrees of freedom. If all physical knowledge could be quantified, it would be possible to construct a reduced model with a DNN, but this is currently an impractical assumption. Therefore, it is difficult for a machine learning approach to achieve the same function as a physicist, who can flexibly interpret phenomena by utilizing explicit or implicit physical knowledge and construct a reduced model.

To overcome this problem, it is usefull to employ methods to extract symmetries of the dynamics system directly from physical data without constructing a reduced model [14, 15, 16, 17, 18, 19, 18, 20, 21]. These methods are derived from Noether's theorem [22], which connects the symmetry of the Hamiltonian and the conservation law. For example, as the study most relevant to this study, Liu et al. have been proposed using deep neuralnetworks and symbolic regression [18], and they have achieved quantitative estimation of complex conservation laws as interpretable form of functions. To infer the conservation laws, it is only needed the tangent space of the manifold of the continuous transformation group that corresponds to the symmetry of the system. Therefore, unlike Hamiltonian estimation, conservation law estimation only requires manifold modeling with at most first-order accuracy. This means that the conservation law can be inferred with arbitrary precision by polynomial approximation. A coordinate system can then be selected based on the system's symmetries on the coordinate system. Furthermore, the obtained symmetries information can also help physicists construct a reduced model.

The purpose of this study is to verify whether nonlinear symmetry can be estimated by the method of Mototake et al [19]. They develop a method for inferring the symmetry of a data manifold modeled by a deep autoencoder [23] and determine the conservation laws of the system. This method allows direct visualization of the symmetries captured by the Auto Encoder through sampling. Although the method of Liu et al. [18] can also estimate the conservation laws as interpretable forms of functions corresponding to nonlinear symmetries, the visualization of symmetries should allow the scientists to work their insight from other viewpoints. Such a property of the method is expected to be useful for extracting complex conservation laws corresponding to nonlinear symmetries in an interpretable form to scientists. The method is also capable, in principle, of estimating complex symmetries, such as invariance of the system to non-linear transformations, but no such symmetry estimation was actually carried out in the study [19]. The purpose of this study is to verify whether the method can estimate the symmetries corresponding to non-linear transformations and to propose modifications to the estimation framework needed to do it. Specifically, we attempt to estimate non-linear transformations corresponding to the conservation law of Runge-Lenz vector present in central force systems obeying the inverse square law.

This paper is organized as follows. In Sec. 2, we show the relationship between the symmetry of the time-series dataset distribution and the conservation law using Noether's theorem according to Mototake's paper [19]. In Sec. 3, we describe the proposed procedure of inferring the non-linear symmetry of the time-series data manifold based on the employed methods [19]. In Sec. 4, to confirm the effectiveness of the proposed methods, we apply them to the system conserving the Runge–Lenz vector in a central force system. In Sec. 5, we present a summary and discussion.

## 2 Theory

### 2.1 Noether's theorem

Noether's theorem establishes a deep connection between the continuous symmetries of a Hamiltonian system and the conservation laws that govern it [22]. It is often described in the $(2d + 1)$-dimensional extended phase space $\Gamma \times \mathbb{R}$, $(\boldsymbol{q}, \boldsymbol{p}) := (q_0 = t, q_1, \cdots, q_d, p_1, \cdots, p_d)$. The Noether's theorem can also be described in the $(2d + 2)$-dimensional space $\Gamma \times \mathbb{R} \times \mathbb{R}$, $(q_0 = t, q_1, \cdots, q_d, p_0 = -H, p_1, \cdots, p_d)$. In this study, we describe the Noether's theory in the $(2d + 2)$-dimensional space as follows. Hamiltonian systems in the $(2d + 2)$-dimensional space $\Gamma \times \mathbb{R} \times \mathbb{R}$ are cinsidered, and restrict ourselves to the case where the system's Hamiltonian belongs to a $C^2$ class function $H(\boldsymbol{q}, \boldsymbol{p})$. The Hamiltonian representation of Noether's theorem is described as follows [24]. Assume that $H(\boldsymbol{q}, \boldsymbol{p})$ and the canonical equations of motion $\frac{\partial H(\boldsymbol{q}, \boldsymbol{p})}{\partial q_i} = -\dot{p}_i$ and $\frac{\partial H(\boldsymbol{q}, \boldsymbol{p})}{\partial p_i} = \dot{q}_i$ are invariant under the infinitesimal transformation $(q_i', p_i') = (q_i + \delta q_{ij}, p_i + \delta p_{ij})$, where $i = 1, \ldots, d$, and $j$ is the index of the direction of the infinitesimal transformation corresponding to a conservation law. Then, on the basis of Noether's theorem, the conserved value $G_j$ satisfies the following equation: $(\delta q_{ij}, \delta p_{ij}) = \left( \frac{\partial G_j}{\partial p_i}, -\frac{\partial G_j}{\partial q_i} \right)$. The canonical transformation that makes the Hamiltonian system invariant is given as

$$\mathfrak{c}_{\text{inv}}(\boldsymbol{\theta}) : \Gamma \times \mathbb{R} \times \mathbb{R} \longrightarrow \Gamma \times \mathbb{R} \times \mathbb{R}, \tag{1}$$

$$(\boldsymbol{q}, \boldsymbol{p}) \longmapsto (Q, \mathcal{P}) := (Q(\boldsymbol{q}, \boldsymbol{p}, \boldsymbol{\theta}), \mathcal{P}(\boldsymbol{q}, \boldsymbol{p}, \boldsymbol{\theta})), \tag{2}$$

where $Q(\boldsymbol{q}, \boldsymbol{p}, \boldsymbol{\theta})$ and $\mathcal{P}(\boldsymbol{q}, \boldsymbol{p}, \boldsymbol{\theta})$ represent the invariant transformation functions of coordinate $(\boldsymbol{q}, \boldsymbol{p})$ to $(Q, \mathcal{P})$, and $\boldsymbol{\theta}$ represents a $d_\theta$-dimensional continuous parameter characterizing transformation that satisfies $Q\left(\boldsymbol{q}, \boldsymbol{p}, \boldsymbol{\theta} = \vec{0}\right) = \boldsymbol{q}$, and $\mathcal{P}\left(\boldsymbol{q}, \boldsymbol{p}, \boldsymbol{\theta} = \vec{0}\right) = \boldsymbol{p}$. In this paper, this transformation is called an invariant transformation. A set of the invariant transformations characterized by the continuous parameters $\boldsymbol{\theta}$ forms a Lie group. By the first-order Taylor expansion of $Q_i(\boldsymbol{q}, \boldsymbol{p}, \boldsymbol{\theta})$ and $\mathcal{P}_i(\boldsymbol{q}, \boldsymbol{p}, \boldsymbol{\theta})$ around $\boldsymbol{\theta} = \vec{0}$, we have the infinitesimal transformation, $(\delta q_{ij}, \delta p_{ij}) = \left( \varepsilon \left. \frac{\partial Q_i(\boldsymbol{q}, \boldsymbol{p}, \boldsymbol{\theta})}{\partial \theta_j} \right|_{\boldsymbol{\theta} = \vec{0}}, \varepsilon \left. \frac{\partial \mathcal{P}_i(\boldsymbol{q}, \boldsymbol{p}, \boldsymbol{\theta})}{\partial \theta_j} \right|_{\boldsymbol{\theta} = \vec{0}} \right)$, where $|\varepsilon| \ll 1$.

### 2.2 Noether's theorem and time-series dataset

In previous study[19], we found that the candidate transformations that make the Hamiltonian and canonical equations invariant are obtained as the transformations that make the subspace

$$S_i := \left\{ \boldsymbol{q}_{t+\Delta t}, \boldsymbol{p}_{t+\Delta t}, \boldsymbol{q}_t, \boldsymbol{p}_t \, \middle| \, H(\boldsymbol{q}_t, \boldsymbol{p}_t) = E_i, \boldsymbol{p}_{t+\Delta t} = \boldsymbol{p}_t - \frac{\partial H(\boldsymbol{q}_t, \boldsymbol{p}_t)}{\partial \boldsymbol{q}_t}, \boldsymbol{q}_{t+\Delta t} = \boldsymbol{q}_t + \frac{\partial H(\boldsymbol{q}_t, \boldsymbol{p}_t)}{\partial \boldsymbol{p}_t} \right\} \tag{3}$$

invariant. We also found taht $S_i$ is understood as a differentiable manifold[19]. Interpolation of differentiable manifolds can be realized by machine learning methods such as deep learning [25, 23, 26, 27, 28, 29]. In the framework, $S_i$ is estimated from a finite number of data $D$ using a deep learning technique.

### 2.3 DNN and data manifold

As mentioned in Sec. 2.2, the subspace $S_i$ could be modeled as a differentiable manifold using a DNN model. In this paper, we refer to such a differentiable manifold as a data manifold.

We explain how a DNN models a $d_m$-dimensional manifold in $d_{\text{in}}$-dimensional space $\boldsymbol{x}$ using one of the simplest DNNs: a feed forward three-layer DNN, for which the input has $d_{\text{in}}$ dimensions, the hidden layer has $d_{\text{h}}(> d_{\text{in}})$ dimensions, and the output has $d_{\text{out}}(< d_{\text{in}}) = d_m$ dimensions. The mapping function $\boldsymbol{f}_{\text{DNN}}(\boldsymbol{x}) = [f_1(\boldsymbol{x}), f_2(\boldsymbol{x}), \cdots, f_{d_{\text{out}}}(\boldsymbol{x})]$ of the DNN is defined as $\boldsymbol{f}_{\text{DNN}}(\boldsymbol{x}) = \boldsymbol{w}^h \boldsymbol{h} = \boldsymbol{w}^h \boldsymbol{\varphi}(\boldsymbol{w}^{\text{in}} \boldsymbol{x})$, where $\boldsymbol{h} = (h_1, h_2, \cdots, h_{d_h})$ is the $d_h$-dimensional output of the hidden layer. We define $\boldsymbol{\varphi}(\cdot)$ as $\boldsymbol{\varphi}(\boldsymbol{w}^{\text{in}} \boldsymbol{x}) = (\varphi_1, \varphi_2, \cdots, \varphi_{d_h}), \varphi_j = \varphi\left[ \sum_i^{d_{\text{in}}} \left( w_{ij}^{\text{in}} x_i \right) \right]$, where $\varphi$ is the activation function. Usually, a sigmoid or ReLU function is used as the activation function. These activation functions are constructed using linear and flat domains. On the basis of these properties of activation functions, $\varphi_j$ maps the input subspace related to the linear domain of the activation function to a one-dimensional space to align the vector $(w_{0j}, w_{1j} \cdots, w_{d_{\text{in}} j})$. If the number of $\varphi_j$ sharing the same input subspace is $d_{\text{out}}$, the $\varphi_j$ defines a $d_{\text{out}}$-dimensional sub-hyperplane. The DNN models the data distribution by continuously pasting these sub-hyperplanes as if they were the tangent spaces of a data manifold.

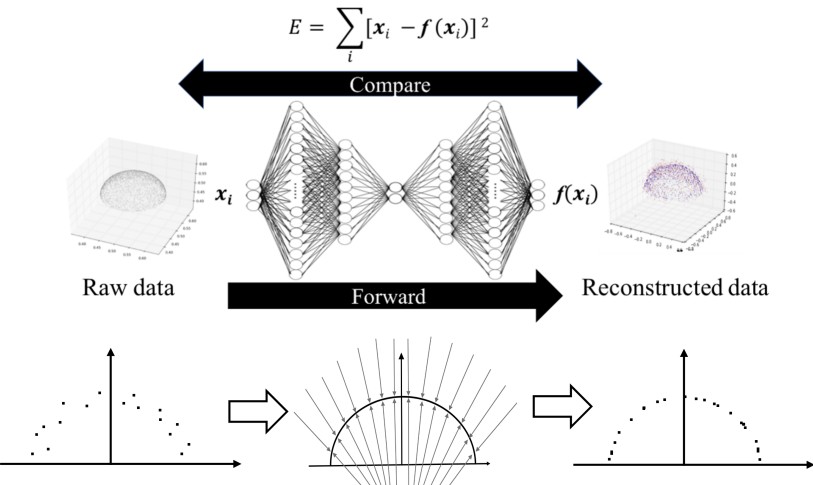

Figure 1: Schematic diagram of method of extracting invariant transformation using autoencoder. Lower panel shows the schematic diagram of the mapping structure of a two-dimensional input space in a DNN trained with data distributed on a black curve. The arrows indicate the compression direction of the input space in the mapping from the input to the hidden layer.

That is, the DNN embeds the input space in the output space by pasting the sub-hyperplanes and compresses the tangent direction of these sub-hyperplanes (Fig. 1). Deeper and more complex DNNs can be understood as a collection of such three-layer DNN. Thus, such deeper DNNs can model more complex manifold structures as a combination of simple manifold structures modeled by a three-layer DNN [27]. Note that the output of a three-layer DNN, a part of the deeper DNN, is referred to as a hidden layer. This is only one example of how a DNN models a data manifold. However, many studies have suggested that there are resemble property in successful trained DNNs [25, 23, 26, 27, 28, 29]. By replacing the input space from $x$ to $\Gamma \times \mathbb{R} \times \mathbb{R}$, we can also model a time-series data manifold $S_i$ using DNN.

In the employed method [19], using a trained DNN that models a time-series data manifold $S_i$, we propose a method of extracting information about the symmetry of a dynamical system. The framework does not require special DNNs, so we can directly utilize the vast knowledge obtained from studies on physical data analysis using DNNs.

## 3 Method

In this section, we describe the employed framework[19] for estimating the symmetry of a time-series dataset of dynamics.

### 3.1 Estimating method of nonlinear symmetry

On the basis of the theory of the relationship between the symmetry of the time-series dataset distribution and the conservation law (Sec. 2.2), we prviously proposed a method[19] of inferring the symmetry of data manifold using the Monte Carlo sampling method. In this study, we extended the methods to extract the symmetry for non-linear transformations. In this section, the symmetry estimation framework is described, together with the extensions for nonlinear-symmetry estimation.

It can be inferred from the discussion in Sec. 2 that data points that are not on the manifold in the input space are attracted to the manifold (Fig. 1). Once the data points are attracted to the manifold in the hidden layer, they continue to exist on the manifold in the output $f(\mathbf{x})$. We propose a method based on this property of DNNs for extracting the symmetry of the data manifold using a deep autoencoder [23]. The deep autoencoder is a model that compresses the input space to a low-dimensional hidden layer and decompresses the layer to an output space with the same dimension as the input space. In the decompression process, only the subspace of the input space around the data manifold is recovered

because of the DNN property. On the basis of this property, we can evaluate whether a transformation $X(\cdot)$ causes the dataset distribution $\{x_i\}_{i=1}^N$ to remain in the same subspace of the data manifold (Fig. 1). The procedure is as follows. First, we train the deep autoencoder using $\{x_i\}_{i=1}^N$ as a training dataset. Second, we input the transformed dataset $\{X(x_i)\}_{i=1}^N$ into the trained deep autoencoder. Note that the deep autoencoder is not trained on the transformed dataset. Third, we evaluate the transformation $X(\cdot)$ using the mean squared error between the input distribution of the dataset and its mapped distribution:

$$E_{\text{samp}}[X(\cdot)] = \frac{1}{N} \sum_{i=1}^N \{X(x_i) - f_{\text{DNN}}[X(x_i)]\}^2 . \tag{4}$$

A smaller $E_{\text{samp}}$ value implies that $X(\cdot)$ is a more invariant transformation. Using the criterion $E_{\text{samp}}$, we approximate the invariant transformation set as $\left\{ X(\cdot) \middle| \arg\min_X E_{\text{samp}}[X(\cdot)] \right\}$, where $\{x_i\}_{i=1}^N$ is $D = \left\{ q_{t_i}^i, p_{t_i}^i, q_{t_i+\Delta t}^i, p_{t_i+\Delta t}^i \right\}_{i=1}^N$, dataset $D$ is generated from dynamics data at energy $E_i$, and $X(\cdot)$ for the transformation $\mathbb{c} : (Q(\cdot, \cdot), P(\cdot, \cdot))$.

To infer the conservation law, it is necessary to estimate the invariant transformation set $M_{\text{invariant}}$ of the manifold $S_i$. The invariant transformation set $M_{\text{invariant}}$ is defined as $M_{\text{invariant}} := \left\{ Q^{S_i}(\cdot, \cdot, \theta), \mathcal{P}^{S_i}(\cdot, \cdot, \theta) \middle| \theta \right\}$. Because $Q^{S_i}(\cdot, \cdot; \theta)$, $\mathcal{P}^{S_i}(\cdot, \cdot; \theta)$ are usually unknown, we infer them to be a subset of a parametric function set $\left\{ Q(\cdot, \cdot; a), P(\cdot, \cdot; a) \middle| a \in \mathbb{R}^{d_a} \right\}$, where $d_a \geq d_\theta$. This function can be complex enough to contain a true transformation function, but it will be more difficult to determine the subset from the finite data. Moreover, significant difficulties arise when estimating invariant infinitesimal transformations. This will be discussed further in the section (Sec. 3.2).

The subset of the true transformation function $M_{\text{invariant}}$ is identified using the trained DNN as

$$M_{\text{invariant}} \sim \left\{ Q(\cdot, \cdot; a), P(\cdot, \cdot; a) \middle| \arg\min_a E_{\text{samp}}[Q(\cdot, \cdot; a), P(\cdot, \cdot; a)] \right\}, \tag{5}$$

$$E_{\text{samp}}[Q(\cdot, \cdot; a), P(\cdot, \cdot; a)] = \frac{1}{N} \sum_{i=1}^N \{[Q(\cdot, \cdot; a), P(\cdot, \cdot; a)] - f_{\text{DNN}}[Q(\cdot, \cdot; a), P(\cdot, \cdot; a)]\}^2 . \tag{6}$$

Next, the invariant transformation is obtained by sampling an element $a_j$ of the parameter vector $a$ following the probability distribution, as in the matrix transformation case

$$\mathrm{P}(a_1, a_2, a_3, \cdots, a_{d_a}) = \frac{1}{Z} \exp\left\{ -\frac{N}{2\sigma^2} E_{\text{samp}}[Q(\cdot, \cdot; a), P(\cdot, \cdot; a)] \right\}. \tag{7}$$

To perform this sampling, we need to specify $\sigma$. Ideally, $\sigma$ should be set to 0. However, it is necessary to set $\sigma$ to an appropriate finite value because errors are included in the time-series dataset and the training results of DNN. Such $\sigma$ affected by noise cannot be set in advance. In addition, the target distributions in this study are assumed to be the global flat minima, because the same $E_{\text{samp}}$ surface following the invariant transformation exists. Generally, such a target distribution needs an enormous amount of time to sample. Therefore, in this study, we use the replica-exchange Monte Carlo (REMC) method [30] as a sampling method to overcome these problems. Such a method enables us to perform efficient sampling by parallel sampling with different noise intensities of $\sigma$ while exchanging noise intensities with each other. In the state of a large noise, we can realize global sampling from the abstract distribution. By exchanging this sampling information with the state of a small noise, we can perform efficient sampling from the target distribution. The procedure of method is summarized in Algorithm 1 of Appendix A.

## 3.2 Estimating method of infinitesimal transformation for nonlinear symmetry

From the $N_a$ sampling results of Eq. (7), $D_a := \{(a_1, a_2 \cdots a_{d_a})_{n_a}\}_{n_a=1}^{N_a}$, the infinitesimal transformations are estimated as follows.

Assuming that $a$ is a differentiable function of $\theta$: $a(\theta)$, $\mathbb{R}^{d_\theta} \to \mathbb{R}^{d_a}$, we can estimate $M_{\text{invariant}}$ as

$$M_{\text{invariant}} = \left\{ Q(\cdot, \cdot; a(\theta)), P(\cdot, \cdot; a(\theta)) \middle| \theta \in \mathbb{R}^{d_\theta} \right\}. \tag{8}$$

The set of invariant transformations $M_{\text{invariant}}$ forms a Lie group, as we mentioned in Sec. 2.1. Therefore, $M_{\text{invariant}}$ constructs a $d_\theta$-dimensional differential manifold in the coordinate space of $\boldsymbol{\theta}$. The infinitesimal transformation is estimated as the tangent vector of the manifold at $\boldsymbol{\theta} = \boldsymbol{0}$ as follows:

$$(\delta\boldsymbol{q}_l, \delta\boldsymbol{p}_l) = \varepsilon\left( \left.\frac{\partial \boldsymbol{Q}(\boldsymbol{q},\boldsymbol{p};\boldsymbol{a}(\theta_l))}{\partial \theta_l}\right|_{\theta_l=0}, \left.\frac{\partial \boldsymbol{P}(\boldsymbol{q},\boldsymbol{p};\boldsymbol{a}(\theta_l))}{\partial \theta_l}\right|_{\theta_l=0} \right). \tag{9}$$

Because $\boldsymbol{a}$ is a differentiable function of $\boldsymbol{\theta}$, the tangent vector is given as

$$(\delta\boldsymbol{q}_l, \delta\boldsymbol{p}_l) = \varepsilon\left( \left.\sum_{k=1}^{d_a}\frac{\partial \boldsymbol{Q}(\boldsymbol{q},\boldsymbol{p};\boldsymbol{a})}{\partial a_k}\frac{\partial a_k(\boldsymbol{\theta})}{\partial \theta_l}\right|_{\boldsymbol{\theta}=\boldsymbol{0}}, \left.\sum_{k=1}^{d_a}\frac{\partial \boldsymbol{P}(\boldsymbol{q},\boldsymbol{p};\boldsymbol{a})}{\partial a_k}\frac{\partial a_k(\boldsymbol{\theta})}{\partial \theta_l}\right|_{\boldsymbol{\theta}=\boldsymbol{0}} \right). \tag{10}$$

Because functions $\boldsymbol{Q}$ and $\boldsymbol{P}$ are defined explicitly, their derivations, $\frac{\partial \boldsymbol{Q}(\boldsymbol{q},\boldsymbol{p};\boldsymbol{a})}{\partial a_k}$ and $\frac{\partial \boldsymbol{P}(\boldsymbol{q},\boldsymbol{p};\boldsymbol{a})}{\partial a_k}$, can be obtained analytically. Therefore, we should only estimate $\left.\frac{\partial a_k(\boldsymbol{\theta})}{\partial \theta_l}\right|_{\boldsymbol{\theta}=\boldsymbol{0}}$ to obtain the infinitesimal transformation.

Because $\boldsymbol{a}(\boldsymbol{\theta})$ is defined as a differentiable function, set $\{\boldsymbol{a}|\boldsymbol{\theta}\in\mathbb{R}^{d_\theta}\}$ constructs a $d_\theta$-dimensional manifold structure in coordinate space $\boldsymbol{a}$. The implicit function representation of the manifold is defined as

$$\begin{cases} f_1(a_1,\cdots,a_{d_a}) = 0 \\ \quad\vdots \\ f_{d_a-d_\theta}(a_1,\cdots,a_{d_a}) = 0 \end{cases}. \tag{11}$$

The Jacobian matrix of $f_k$ for the parameters of subset $\boldsymbol{a}$, $(b_1, b_2, \cdots, b_{d_\theta}) \subset \boldsymbol{a}$, is defined as $J_{kl} = \frac{\partial f_k(a_1,\cdots,a_{d_a})}{\partial b_l}$. If the Jacobian matrix at $\boldsymbol{a}_{\text{id}}$ becomes nonsingular, from the implicit function theorem, variables other than $(b_1, b_2, \cdots, b_{d_\theta})$, $\{c_k\}_{k=1}^{d_a-d_\theta} := A' \setminus \{b_l\}_{l=1}^{d_\theta}$, can be expressed as $c_k = g_i(b_1, \cdots, b_{d_\theta})$. This means that $\boldsymbol{\theta}$ can be replaced by $\boldsymbol{b}$. In this case, $\left.\frac{\partial a_k(\boldsymbol{\theta})}{\partial \theta_l}\right|_{\boldsymbol{\theta}=\boldsymbol{0}}$ is estimated as the tangent vector $\left.\frac{\partial a_k(\boldsymbol{b})}{\partial b_l}\right|_{\boldsymbol{a}=\boldsymbol{a}_{\text{id}}}$ at identity map $\boldsymbol{a}_{\text{id}} \in \{\boldsymbol{a}|\boldsymbol{Q}(\cdot,\cdot;\boldsymbol{a})=\boldsymbol{q}, \boldsymbol{P}(\cdot,\cdot;\boldsymbol{a})=\boldsymbol{p}\}$. This implies that, around $e_I$, the implicit equations in Eq. (11) representing the manifold $M_{\text{invariant}}$ can be decomposed into the following $d' - d_\theta$ simultaneous equations:

$$\begin{cases} h_1(c_1, b_1, \cdots, b_{d_\theta}) = 0 \\ \quad\vdots \\ h_{d'-d_\theta}(c_{d'-d_\theta}, b_1, \cdots, b_{d_\theta}) = 0 \end{cases}, \tag{12}$$

where $b_l$ corresponds to the continuous parameter $\theta_l$ of continuous transformation $[\mathcal{Q}(\boldsymbol{q},\boldsymbol{p},\boldsymbol{\theta}), \mathcal{P}(\boldsymbol{q},\boldsymbol{p},\boldsymbol{\theta})]$. Differentiating these equations with respect to $b_l$ around a point $e_I$ yields $d' - d_\theta$ simultaneous partial differential equations,

$$\begin{cases} \frac{\partial}{\partial b_l}h_1(c_1, b_1, \cdots, b_{d_\theta})|_{A'=e_I} = 0 \\ \quad\vdots \\ \frac{\partial}{\partial b_l}h_{d'-d_\theta}(c_{d'-d_\theta}, b_1, \cdots, b_{d_\theta})|_{A'=e_I} = 0 \end{cases}. \tag{13}$$

Solving these simultaneous partial differential equations gives the tangent vector $\left.\frac{\partial a(b_l)}{\partial b_l}\right|_{\boldsymbol{a}=\boldsymbol{a}_{\text{id}}}$ of the manifold at $\boldsymbol{a}_{\text{id}}$. Thus, if $h_k$ can be regressed with the sampling result $D_a$ as the polynomial of $\{b_l\}_{l=1}^{d_\theta}$, the conservation law can be inferred. Thus, we can estimate the infinitesimal transformation $(\delta\boldsymbol{q}_l, \delta\boldsymbol{p}_l)$ from the sampling result $D_a$. Thus, in principle, the previously proposed method can be applied to general coordinate transformations including nonlinear transformation. But, to estimate interpretable conservation laws, we would need to model nonlinear transformations of appropriate complexity as parametric functions. This is as difficult as setting up a reduced coordinate system.

## 3.3 Runge Lenz vector and nonlinear transformation

From the discussion in the Sec. 3.1 and 3.2, in order to search for the non-linear symmetries required for conservation law estimation, it is necessary to set up a parametric function that can represent

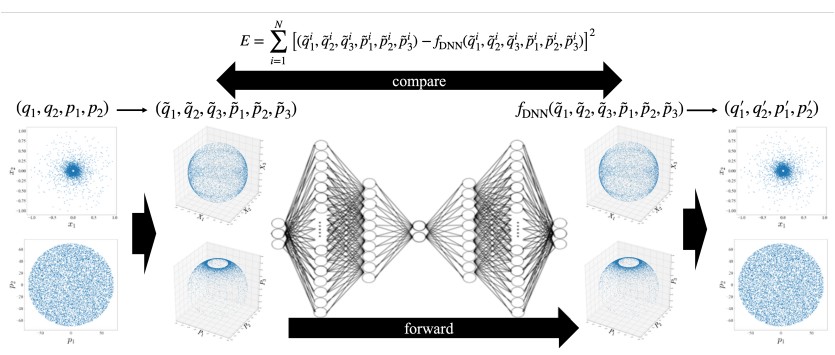

$$E = \sum_{i=1}^{N} \left[ (\tilde{q}_1^i, \tilde{q}_2^i, \tilde{q}_3^i, \tilde{p}_1^i, \tilde{p}_2^i, \tilde{p}_3^i) - f_{\text{DNN}}(\tilde{q}_1^i, \tilde{q}_2^i, \tilde{q}_3^i, \tilde{p}_1^i, \tilde{p}_2^i, \tilde{p}_3^i) \right]^2$$

compare

$$(q_1, q_2, p_1, p_2) \longrightarrow (\tilde{q}_1, \tilde{q}_2, \tilde{q}_3, \tilde{p}_1, \tilde{p}_2, \tilde{p}_3) \qquad f_{\text{DNN}}(\tilde{q}_1, \tilde{q}_2, \tilde{q}_3, \tilde{p}_1, \tilde{p}_2, \tilde{p}_3) \longrightarrow (q_1', q_2', p_1', p_2')$$

forward

Figure 2: Schematic diagram of proposed framework.

the non-linear transformation to be estimated. On the other hand, it is generally difficult to pre-set such parametric functions. This difficulty could be overcome by finding a class of parametric functions to explore that can be used generically in certain domains based on physical knowledge. The purpose of this study is therefore to explore a class of parametric functions for such non-linear transformations through the estimation of non-linear transformations corresponding to the Runge Lenz vector, which is a hidden conservation law for central force potential systems where the force is inversely proportional to the square of the radius.

$$H_3 = \frac{1}{2m}\boldsymbol{p}^2 + G\frac{mM}{|\boldsymbol{q}|} \tag{14}$$

First, we describe the geometrical structure of the symmetry of the Runge Lenz vector following previous studies [31, 32]. Consider the motion of the central force potential in six-dimensional phase space: $(\boldsymbol{q}, \boldsymbol{p}) = (q_1, q_2, q_3, p_1, p_2, p_3)$. In this system, the Laplace–Runge–Lenz vector, $\vec{A} = \boldsymbol{p} \times L - mG\frac{\boldsymbol{q}}{\|\boldsymbol{q}\|_2}$, $L = \boldsymbol{q} \times \boldsymbol{p}$, is conserved. The Runge Lenz vector corresponds to the SO(4) symmetry in the coordinate space $(\tilde{\boldsymbol{q}}, \tilde{q}_4, \tilde{\boldsymbol{p}}, \tilde{p}_4) = (\tilde{q}_1, \tilde{q}_2, \tilde{q}_3, \tilde{q}_4, \tilde{p}_1, \tilde{p}_2, \tilde{p}_3, \tilde{p}_4)$, defined as

$$\tilde{\boldsymbol{q}} = \tilde{\boldsymbol{q}}(\boldsymbol{q}, \boldsymbol{p}) := \frac{\boldsymbol{q}}{\|\boldsymbol{q}\|_2} - \frac{\boldsymbol{q} \cdot \boldsymbol{p}}{mG}\boldsymbol{p}, \quad \tilde{q}_4 = \tilde{q}_4(\boldsymbol{q}, \boldsymbol{p}) := \frac{p_0}{mG}\boldsymbol{q} \cdot \boldsymbol{p}, \tag{15}$$

$$\tilde{\boldsymbol{p}} = \tilde{\boldsymbol{p}}(\boldsymbol{q}, \boldsymbol{p}) := \frac{2p_0\boldsymbol{p}}{p_0^2 + \boldsymbol{p}^2}, \quad \tilde{p}_4 = \tilde{p}_4(\boldsymbol{q}, \boldsymbol{p}) := \frac{\boldsymbol{p}^2 - p_0^2}{p_0^2 + \boldsymbol{p}^2}, \tag{16}$$

where $p_0 = \sqrt{-2mE}$. The transformed coordinate satisfies the conditions $\tilde{\boldsymbol{q}}^2 + \tilde{q}_4^2 = 1$, $\tilde{\boldsymbol{p}}^2 + \tilde{p}_4^2 = 1$, and $\tilde{\boldsymbol{q}} \cdot \tilde{\boldsymbol{p}} + \tilde{q}_4\tilde{p}_4 = 0$. Let us assume that the matrix representation of SO(4) is given by $A$. Moreover, assume the transformation is represented as $\tilde{\boldsymbol{q}}'^T = A\tilde{\boldsymbol{q}}^T$ and $\tilde{\boldsymbol{p}}'^T = A\tilde{\boldsymbol{p}}^T$.

We investigate the correspondence between the $4 \times 4$ matrix representation $A$ of the SO(4) symmetry in $(\tilde{\boldsymbol{q}}, \tilde{\boldsymbol{p}})$ space and the coordinate transformation in $(\boldsymbol{q}, \boldsymbol{p})$ space. Because the inverse of the coordinate transformation is given by

$$\boldsymbol{q} = \boldsymbol{q}(\tilde{\boldsymbol{q}}, \tilde{q}_4, \tilde{\boldsymbol{p}}, \tilde{p}_4) = -\frac{G}{2E}[(1 - \tilde{p}_4)\tilde{\boldsymbol{q}} + \tilde{q}_4\tilde{\boldsymbol{p}}], \quad \boldsymbol{p} = \boldsymbol{p}(\tilde{\boldsymbol{q}}, \tilde{q}_4, \tilde{\boldsymbol{p}}, \tilde{p}_4) = \sqrt{-2mE}\frac{\tilde{\boldsymbol{p}}}{1 - \tilde{p}_4}, \tag{17}$$

the transformation of SO(4) in the original space becomes

$$\boldsymbol{Q}(\tilde{\boldsymbol{q}}, \tilde{q}_4, \tilde{\boldsymbol{p}}, \tilde{p}_4) = \boldsymbol{q}(\tilde{\boldsymbol{Q}}, \tilde{Q}_4, \tilde{\boldsymbol{P}}, \tilde{P}_4), \quad \boldsymbol{P}(\tilde{\boldsymbol{q}}, \tilde{q}_4, \tilde{\boldsymbol{p}}, \tilde{p}_4) = \boldsymbol{p}(\tilde{\boldsymbol{Q}}, \tilde{Q}_4, \tilde{\boldsymbol{P}}, \tilde{P}_4), \tag{18}$$

$$\begin{pmatrix} \tilde{\boldsymbol{Q}}^t \\ \tilde{Q}_4 \end{pmatrix} = A\begin{pmatrix} \tilde{\boldsymbol{q}}^t \\ \tilde{q}_4 \end{pmatrix}, \quad \begin{pmatrix} \tilde{\boldsymbol{P}}^t \\ \tilde{P}_4 \end{pmatrix} = A\begin{pmatrix} \tilde{\boldsymbol{p}}^t \\ \tilde{p}_4 \end{pmatrix}. \tag{19}$$

Thus, the Runge Lenz vector has linear symmetry in the space beyond which it maps the phase space with certain non-linear transformations. Such symmetry estimates suggest that it is useful to assume a class of non-linear transformations, such as stereo mapping, as a class of mapping transformations of phase space.

In this study, we propose a framework in which the non-linear symmetry is assumed to be a combination of a coordinate transformation and a linear transformation (Fig. 2), each of which is estimated

independently of the other. A machine learning framework to estimate nonlinear symmetries has been already proposed using symbolic regression [18], they can also estimate the conservation laws as interpretable forms of functions corresponding to nonlinear symmetries. A method has also been proposed [21] to visualize conservation laws in the space in which they are embedded. The advantage of our method is to allow direct visualization of the manifolds formed by Lie groups. It should allow the scientists to work their insight from other viewpoints.

In this study, we check whether it is possible to estimate the linear symmetry corresponding to the Runge Lenz vector in the space of its mapping destination when the previously mentioned coordinate transformations are known. It is not obvious that the estimation will work even when the coordinate transformations are known. That is, under a non-linear coordinate transformation, the measure changes from a point in the original space to a point in mapped space, and if the data are finite, even if the data manifold has a uniform density in the original space, there will be regions where the density is almost zero at the mapping destination (see Fig. 2). This makes it difficult to estimate symmetry.

## 4 Results

We applied the proposed method to a system of central force potentials (Eq. 14). Specifically, for simulation data generated at all energies and initial conditions under the Hamiltonian of the central force potential (Eq. 14), an estimation of the set of transformations that make the data manifold invariant was performed in the framework of the following linear transformation after applying a coordinate transformation [Eqs.(23) and (24)]:

$$\begin{pmatrix} \tilde{Q}_1 \\ \tilde{Q}_2 \\ \tilde{Q}_3 \\ \tilde{P}_1 \\ \tilde{P}_2 \\ \tilde{P}_3 \end{pmatrix} = \begin{pmatrix} a_{11}, a_{12}, & 0 &, & 0 &, & 0 &, & 0 \\ a_{21}, a_{22}, & 0 &, & 0 &, & 0 &, & 0 \\ 0 &, & 0 &, & 1 &, & 0 &, & 0 &, & 0 \\ 0 &, & 0 &, & 0 &, a_{11}, a_{12}, & 0 \\ 0 &, & 0 &, & 0 &, a_{21}, a_{22}, & 0 \\ 0 &, & 0 &, & 0 &, & 0 &, & 0 &, & 1 \end{pmatrix} \begin{pmatrix} \tilde{q}_1 \\ \tilde{q}_2 \\ \tilde{q}_3 \\ \tilde{p}_1 \\ \tilde{p}_2 \\ \tilde{p}_3 \end{pmatrix} \tag{20}$$

The estimation results of the proposed method confirm that a set of target transformations corresponding to the Lungerenz vector can be obtained (Fig. 3). Specifically, for the matrix elements $a_{11}$ and $a_{12}$ corresponding to cos and sin, a set of circular symmetric transformations was obtained, and for the matrix elements $a_{11}$ and $a_{22}$ corresponding to cos and cos, a set of diagonal symmetric transformations (Fig. 3).

## 5 Summary and Discussion

This study suggests that the employed method [19] of directly visualizing manifolds formed by Lie algebras is also effective for non-linear transformations, by separating the transformation function for verifying symmetry into a coordinate transformation and a linear transformation. In this study, it was confirmed that linear transformations can be estimated under known coordinate transformations. As a result, we succeeded in extracting a set of symmetric transformations, despite the fact that the nonlinear coordinate transformations resulted in large differences in measures between the original and mapped spaces.

In the future, we will further attempt to estimate the non-linear coordinate transformations and estimate the conserved values based on them. It is necessary to express non-linear mapping transformations in terms of parametric functions, in which case it may be useful to use a function class of stereo mapping, such as the one used in this study. It is then necessary to represent the non-linear coordinate transformations by parametric functions. The results of this study suggest that it is useful to use a function class of stereo mapping, such as the one used in this study, as its parametric function.

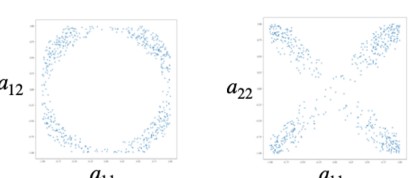

Figure 3: Estimation results of symmetric transformation set corresponding to Runge Lentz vector.

## Acknowledgements

We would like to thank the all reviewers for their patience and kind comments on our manuscript, which was incomplete, especially in that the description of important previous studies was missing. This work was supported by JST, PRESTO Grant Number JPMJPR212A and JSPS KAKENHI 22K13979, 23H03460.

## Appendix A

The procedure of proposed method is summarized in Algorithm 1.

---

**Algorithm 1** Estimation of the invariant transformation set [19]

---

**Input**: dataset $D = \left\{ \boldsymbol{q}^i_{t_i}, \boldsymbol{p}^i_{t_i}, \boldsymbol{q}^i_{t_i+\Delta t}, \boldsymbol{p}^i_{t_i+\Delta t} \right\}^N_{i=1}$ in a given coordinate system.

**Output**: Invariant transformation set $D_a = \{(a_1, a_2, a_3, \cdots, a_{d_a})_{n_a}\}^{N_a}_{n_a=1}$.

**Step 1**: Train the deep autoencoder with dataset $D$.

**Step 2**: Using the trained deep autoencoder and REMC method, sampling transformation parameters $a_1, a_2, a_3, \cdots, a_{d_a}$ from multiple probability distributions $P'(a_1, a_2, a_3, \cdots, a_{d_a})$ corresponding to different noise intensities $\sigma'$.

**Step 3**: Select $\sigma'$ from the distribution structure of the sampling results and output the sampling result of the selected $\sigma'$ state as $D_a$.

---

## References

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
