# OpenReview forum: "Extracting Nonlinear Symmetries From Trained Neural Networks on Dynamics Data"
_NeurIPS.cc/2023/Workshop/AI4Science — NeurIPS2023-AI4Science Poster_

### Official Review · Reviewer_iL7S · 2023-10-10
**Limited experiments, no comparisons, limited applicability, and not properly anonymized**

**Rating:** 4
**Confidence:** 4

**Review:**

This work proposes a method for identifying symmetries and conservation laws from dynamics data.

# Limited experiments
The authors show only a single experiment with very little detail as to how it was performed or how to interpret the quality of the result.

1. Is equation 20 given to the method as the parameterization for the symmetry or is it discovered?
2. In such a system with multiple conservation laws, how does the method separately identify each symmetry?

# No comparisons
This is an interesting area with many prior works. Unfortunately, the authors fail to discuss or compare with prior methods for discovering conservation laws [1–6].

1. E. Kaiser, J. N. Kutz, and S. L. Brunton, Discovering Conservation Laws from Data for Control, in 2018 IEEE Conference on Decision and Control (CDC) (IEEE, 2018), pp. 6415–6421.

2. P. Y. Lu, R. Dangovski, and M. Soljačić, Discovering Conservation Laws Using Optimal Transport and Manifold Learning, Nat. Commun. 14, 4744 (2023).

3. Z. Liu, V. Madhavan, and M. Tegmark, Machine Learning Conservation Laws from Differential Equations, Phys. Rev. E. 106, 045307 (2022).

4. Z. Liu and M. Tegmark, Machine Learning Conservation Laws from Trajectories, Phys. Rev. Lett. 126, 180604 (2021).

5. S. Ha and H. Jeong, Discovering Invariants via Machine Learning, Phys. Rev. Res. 3, (2021).

6. S. J. Wetzel, R. G. Melko, J. Scott, M. Panju, and V. Ganesh, Discovering Symmetry Invariants and Conserved Quantities by Interpreting Siamese Neural Networks, Phys. Rev. Res. 2, 033499 (2020).

# Limited applicability
1. Despite the title emphasizing "nonlinear symmetries", the proposed approach only appears to be applicable to symmetries which are global linear/affine transformations of the chosen coordinates. To get around this issue for the unusual symmetry corresponding to the conserved Runge–Lenz vector, the authors explicitly transform the coordinates into a new set of coordinates for which the symmetry acts linearly. They do not demonstrate how this may be done in a more general setting when the proper transformation is unknown. Unfortunately, this more general setting is precisely where a method for discovering conservation laws would be practically useful.
2. The reliance on an autoencoder's reconstruction accuracy to identify symmetries seems like a rather ad-hoc choice that would be sensitive to the architecture of the autoencoder. Why should we expect that an autoencoder's reconstruction accuracy is meaningful as a signal away from the data manifold? How does the architecture, in particular the size of the smallest middle layer, depend on the dimensionality of the manifold that you are trying to reconstruct? How would I choose this without prior knowledge?

# Not properly anonymized
The authors repeatedly refer to a prior work as "our previously proposed methods" without any attempt at anonymization.

---

### Official Review · Reviewer_kgSB · 2023-10-14
**Extracting Nonlinear Symmetries**

**Rating:** 5
**Confidence:** 3

**Review:**

The paper extends the method in [14] to extracting nonlinear symmetries from dynamics of a physics system. The proposed method estimates invariant transformations from a deep neural network trained on time series data. By separating transformations into a coordinate transformation and a linear transformation, the method is able to extract nonlinear symmetries. Empirical results on a system of central force potentials confirms that the method successfully finds the set of invariant transformations.

### Pros
- Nonlinear symmetry is common in physical systems but has received little attention. The proposed method takes a step towards addressing this gap, and has promising preliminary experimental results.
- The motivation to develop alternative methods to reduced models is explained well in the introduction.

### Cons
-	A few claims in Section 3.1 are not well supported. For example, statements such as “data points that are not on the manifold in the input space are attracted to the manifold” and “In the decompression process, only the subspace of the input space around the data manifold is recovered because of the DNN property” can be made more precise and require either a proof or empirical evidence.
-	Although the proposed method does not need to construct a reduced model, it faces the same challenge of requiring knowledge of physics. In particular, the parametric function class need to be known when finding nonlinear symmetries.
-	There is no convincing evidence that the proposed method is more effective than existing work on extracting symmetry from data, such as [Krish Desai, Benjamin Nachman, and Jesse Thaler. Symmetry discovery with deep learning. Physical Review D, 105(9):096031, 2022.]
-	The system used in the experiment is small, which is acceptable as a proof-of-concept. However, larger scale empirical study may significantly improve the usefulness of the proposed method.
-	The phrase “our previously proposed…” in line 137, 141, and 266 potentially violates anonymity.

---

### Meta-Review · Area_Chair_Mtp9 · 2023-10-26

**Recommendation:** Accept (Poster)
**Confidence:** 3

**Metareview:**

This work proposes a method for estimating invariant transformations from a neural network trained on times series data. The reviewers have raised several concerns that the authors should take seriously. Despite being at an early stage, this work could be of interest to the broader community and would benefit from critical feedback with more interactions. Recommendation: Poster.